# Improving Social Justice in COVID-19 Health Research: Interim Guidelines for Reporting Health Equity in Observational Studies

**DOI:** 10.3390/ijerph18179357

**Published:** 2021-09-04

**Authors:** Alba Antequera, Daeria O. Lawson, Stephen G. Noorduyn, Omar Dewidar, Marc Avey, Zulfiqar A. Bhutta, Catherine Chamberlain, Holly Ellingwood, Damian Francis, Sarah Funnell, Elizabeth Ghogomu, Regina Greer-Smith, Tanya Horsley, Clara Juando-Prats, Janet Jull, Elizabeth Kristjansson, Julian Little, Stuart G. Nicholls, Miriam Nkangu, Mark Petticrew, Gabriel Rada, Anita Rizvi, Larissa Shamseer, Melissa K. Sharp, Janice Tufte, Peter Tugwell, Francisca Verdugo-Paiva, Harry Wang, Xiaoqin Wang, Lawrence Mbuagbaw, Vivian Welch

**Affiliations:** 1Biomedical Research Institute Sant Pau, Hospital de la Santa Creu i Sant Pau, 08025 Barcelona, Spain; 2Department of Health Research Methods, Evidence, and Impact, McMaster University, Hamilton, ON L8S 4L8, Canada; lawsod3@mcmaster.ca (D.O.L.); noordus@mcmaster.ca (S.G.N.); mbuagblc@mcmaster.ca (L.M.); 3Faculty of Medicine, School of Epidemiology and Public Health, University of Ottawa, Ottawa, ON K1G 5Z3, Canada; odewi090@uottawa.ca (O.D.); thorsley@royalcollege.ca (T.H.); jlittle@uottawa.ca (J.L.); mngui058@uottawa.ca (M.N.); ptugwell@uottawa.ca (P.T.); vwelch@campbellcollaboration.org (V.W.); 4Public Health Agency of Canada, Ottawa, ON K1A 0K9, Canada; marc.t.avey@gmail.com; 5Centre for Global Child Health, Hospital for Sick Children, Toronto, ON M5G 1X8, Canada; zulfiqar.bhutta@sickkids.ca; 6Institute for Global Health & Development, The Aga Khan University, Karachi 74800, Pakistan; 7School of Nursing and Midwifery, La Trobe University, Melbourne, VIC 3086, Australia; cacham@unimelb.edu.au; 8Ngangk Yira Research Centre for Aboriginal Health and Social Equity, Murdoch University, Perth, WA 6150, Australia; 9Department of Psychology, Faculty of Arts and Social Sciences, Carleton University, Ottawa, ON K1S 5B6, Canada; holly.ellingwood@carleton.ca; 10Public Safety, Ottawa, ON K1A 0P8, Canada; 11Center for Health and Social Issues, School of Health and Human Performance, Georgia College, Milledgville, GA 31061, USA; damian.francis@gcsu.edu; 12Department of Family Medicine, Queen’s University, Kingston, ON K7L 3G2, Canada; Sarah.Funnell@ottawa.ca; 13Department of Family Medicine, Faculty of Medicine, University of Ottawa, Ottawa, ON K1G 5Z3, Canada; 14Bruyère Research Institute, University of Ottawa, Ottawa, ON K1N 5C8, Canada; etanjongghogomu@bruyere.org; 15Healthcare Research Associates, LLC/The S.T.A.R. Initiative, Los Angeles, CA 90033, USA; healthcareresearch@sbcglobal.net; 16Research Unit, Royal College of Physicians and Surgeons of Canada, Ottawa, ON K1S 5N8, Canada; 17Applied Health Research Center, St. Michael’s Hospital, Toronto, ON M5B 1W8, Canada; clara.juando@utoronto.ca; 18Dalla School of Public Health, University of Toronto, Toronto, ON M5T 3M7, Canada; 19Faculty of Health Sciences, School of Rehabilitation Therapy, Queen’s University, Kingston, ON K7L 3N6, Canada; janet.jull@queensu.ca (J.J.); arizv036@uottawa.ca (A.R.); 20Faculty of Social Sciences, School of Psychology, University of Ottawa, Ottawa, ON K1N 6N5, Canada; kristjan@uottawa.ca; 21Clinical Epidemiology Program, Ottawa Hospital Research Institute, Ottawa, ON K1H 8L6, Canada; snicholls@ohri.ca; 22Faculty of Public Health and Policy, London School of Hygiene & Tropical Medicine, London WC1E 7HT, UK; mark.petticrew@lshtm.ac.uk; 23Epistemonikos Foundation, Santiago 7510299, Chile; radagabriel@epistemonikos.org (G.R.); fverdugo@epistemonikos.org (F.V.-P.); 24UC Evidence Center, Cochrane Chile Associated Center, Pontificia Universidad Católica de Chile, Santiago Región Metropolitana, Santiago 8331150, Chile; 25Knowledge Translation Program, Li Ka Shing Knowledge Institute, Unity Health Toronto, Toronto, ON M5B 1T8, Canada; Larissa.Shamseer@unityhealth.to; 26Health Research Board Centre for Primary Care Research, Department of General Practice, Royal College of Surgeons in Ireland, Dublin DO2 H638, Ireland; melissasharp@rcsi.ie; 27Hassanah Consulting, Seattle, WA 98122, USA; janicetufte@yahoo.com; 28Faculty of Medicine, University of Ottawa, Ottawa, ON K1H 8M5, Canada; harry.wang@uottawa.ca; 29Michael G. DeGroote Institute for Pain Research and Care, McMaster University, Hamilton, ON L8S 4L8, Canada; wangx431@mcmaster.ca

**Keywords:** health inequities, observational studies, COVID-19, guidelines, reporting, public health, vulnerable populations

## Abstract

The COVID-19 pandemic has highlighted the global imperative to address health inequities. Observational studies are a valuable source of evidence for real-world effects and impacts of implementing COVID-19 policies on the redistribution of inequities. We assembled a diverse global multi-disciplinary team to develop interim guidance for improving transparency in reporting health equity in COVID-19 observational studies. We identified 14 areas in the STROBE (Strengthening the Reporting of Observational Studies in Epidemiology) checklist that need additional detail to encourage transparent reporting of health equity. We searched for examples of COVID-19 observational studies that analysed and reported health equity analysis across one or more social determinants of health. We engaged with Indigenous stakeholders and others groups experiencing health inequities to co-produce this guidance and to bring an intersectional lens. Taking health equity and social determinants of health into account contributes to the clinical and epidemiological understanding of the disease, identifying specific needs and supporting decision-making processes. Stakeholders are encouraged to consider using this guidance on observational research to help provide evidence to close the inequitable gaps in health outcomes.

## 1. Introduction

“We are not all in the same boat. We are all in the same storm. Some are on super-yachts. Some have just the one oar” [1]*—*Damian Barr.

The coronavirus disease (COVID-19) public health emergency has exacerbated pre-existing social, political, and economic factors driving health inequities and led to an increase in unjust and avoidable differences in health outcomes [2,3]. Furthermore, the responses to the COVID-19 pandemic from healthcare systems, governments, and supranational actors have raised concerns about differential impacts across population groups [4,5,6]. The complex interplay of socio-political factors related to housing, employment, and public health measures have led to excess risk and burden of COVID-19 among marginalised communities [7,8,9]. For example, local inequalities in income and healthcare resources promote virus propagation [10], and lack of public health capacity in low-income countries hampers global coronavirus tracing [11].

Understanding the differential effects of COVID-19 on health outcomes is a prerequisite to adopting efficient and fair measures within our societies [12]. Evidence shows that an approach informed by an understanding of social determinants of health helps to understand the incidence and outcomes of most diseases and to address the roots of health inequities [13,14,15]. However, data examining the role of the social determinants of health within the context of the COVID-19 pandemic have been less prominent than research focusing on the biomedical paradigm [16,17]. A recent rapid review of COVID-19 studies examining infection, health service use, and health outcomes, found a minority of studies that had assessed outcomes according to social determinants of health [18]. Moreover, social determinants are rarely incorporated into mathematical modelling studies, which have achieved substantive importance in informing policy-makers about the impact of disease and interventions in the population over time [19,20]. Thus, these studies may mask the differential exposure and effects across social determinants that are driving systemic inequalities [21].

Health inequities are differences in health that are avoidable and unfair [22]. Structural and systemic inequities in opportunities for health shape vulnerability, defined as exposure to risk, susceptibility to disease, and capability of individuals and communities to respond [23]. For example, structural racism in medicine continues to have profound and adverse impacts on health equity [24,25]. Moreover, a distinction can be drawn between medical vulnerability (those infected individuals who experience poorer prognosis) and functional vulnerability (those individuals who are more susceptible to infection but do not necessarily have a worse prognosis [26]. Thoughtful reporting on sociodemographic data and discussion of social-ethical issues can help reveal social barriers and facilitators to inform evidence-based health policy [27]. The paucity of research examining the impact of the social determinants of health on COVID-19-relevant outcomes may, in part, be due to a lack of agreed criteria to identify populations experiencing vulnerability and information should be reported. The PROGRESS-Plus framework provides a conceptual and practical framework that researchers can use to improve the reporting of social determinants of health. In short, PROGRESS-Plus is comprised of Place of residence (e.g., urban/rural area, low and middle-income country), Race/ethnicity/culture/language, Occupation, Gender or sex, Religion, Education, Socioeconomic status, Social capital, and other contextual factors that facilitate disadvantage (e.g., disability) [28].

Social groups can experience vulnerability across multiple and intersecting PROGRESS-Plus factors. For example, nursing home staff and home care workers who lack health insurance are at high risk for exposure to coronavirus and differential healthcare access [29]. The PROGRESS-Plus framework is useful to systematically assess, synthesise, and present the evidence on the effectiveness of interventions to reduce differential impacts of exposures and interventions across social groups. For instance, our team has previously developed a conceptual framework for identifying and mitigating inequitable harms of COVID-19 lockdown measures [30].

Observational studies (cohorts, cross-sectional studies, case-control studies) are well-suited to study questions related to understanding health inequities such as access, implementation, and adherence questions [31,32]. While well-conducted randomised clinical trials (RCTs) are often perceived to be the gold standard for assessing the efficacy of an intervention, observational studies provide evidence on long-term effects (e.g., safety) and complementary evidence of real-world effects in which inequities modify community effectiveness or implementation of interventions results in a redistribution of inequities at the population level. In the case of COVID-19, observational studies can provide evidence on the distribution of the uptake of vaccines in real-world settings across social factors such as age, sex, and socioeconomic status as well as population effectiveness across these factors [33]. Observational coronavirus-related investigations can address questions about treatment acceptability, feasibility, or unconscious bias of healthcare providers. Observational research also provides insight into transmission dynamics, diagnosis, and prognosis. Furthermore, observational evaluative studies are well-suited to study COVID-19 policies and other population-level interventions where evaluations capitalise on naturally occurring variations in implementation. Observational research can also be used to study the distribution of societal consequences on mental health, food insecurity, job loss, and healthcare access [34,35,36,37].

The continued public health emergency posed by the pandemic and the unjust distribution of the health and societal burden highlighted the vast inequities in diagnostics and vaccinations as well as solutions, both within and between countries. This motivated our global multi-disciplinary, multi-stakeholder team to develop interim guidance on transparency in assessing health equity in observational studies related to COVID-19 [38,39]. Reporting guidelines are an important tool to reduce research waste and promote transparency [40]. Endorsement of reporting guidelines results in more transparent reporting [41]. Although collecting social data is a prerequisite, reporting them is also essential to enable inequities in clinical and public health decisions to be targeted. We aim to extend the well-known STROBE (Strengthening the Reporting of Observational Studies in Epidemiology) reporting guidelines [42] to enhance transparent reporting of health equity considerations.

Indigenous Peoples globally are a priority population for improving health equity, with long-standing structural inequities and lack of access to essential health determinants, including self-determination and connection to the land [43,44]. We engaged with Indigenous scholars, citizens, and stakeholders to co-produce this guidance and appraise its relevance to research with Indigenous Peoples. Additional groups experiencing different dimensions of health inequities were engaged throughout the project to bring an intersectional lens. This engaged approach to partnership reinforces our aspiration to improve accountability and social justice in observational studies.

## 2. Materials and Methods

To produce this interim guidance, we took the following steps: (1) conceptualization of the project, (2) obtaining funding, (3) registering the protocol of the project, (4) engaging a diverse team of knowledge users and citizens, (5) holding regular meetings, (6) working on interim guidelines iteratively the interim using collaborative online platforms, (7) selecting examples of COVID-19 observational studies examining any PROGRESS-Plus factor, and (8) discussing the draft and the final manuscript.

The STROBE-Equity project is registered in the EQUATOR Network (Enhancing the QUAlity and Transparency Of health Research) [38], co-led by four investigators (SF, JJ, LM, VW) and funded by the Canadian Institutes of Health Research. The project is based on the methods for developing reporting guidelines described by Moher et al. [45], with modifications to include people with lived experience of inequity on the team, conduct an author survey, and seek feedback through qualitative key informant interviews [46] in order to co-produce this research with authentic partnerships that value different types of knowledge and participation. To develop this interim guidance, we formed a core writing team that met weekly and included members with expertise in Indigenous health research (AA, DOL, SGN1, OD, AR, EG, VW, LM, SGN2, LS, MKS, SF, JJ, PT, JL). We reviewed other checklists related to equity in research to develop items for the interim STROBE- Equity guidelines. We reviewed the CONSORT (Consolidated Standards of Reporting Trials)-Equity extension [47] which includes 16 extension items related to formulating equity objectives, describing ethical procedures, reporting equity analyses and interpreting equity findings. We assessed the SAGER (Sex and Gender Equity in Research) reporting guidelines [48] and the PRISMA (Preferred Reporting Items for Systematic Reviews and Meta-Analyses)-Equity guidelines [49], both of which contain items related to evaluating equity in research [39]. We convened a citizen panel (HE, JT, RG) with lived experience of health inequities who met monthly to seek their views and engage them as citizen co-leads in designing the overall project. We then planned this study together, using conference calls, email, and an online text processor for collaborative manuscript editing.

We searched for examples of COVID-19 studies using the L·OVE (Living OVerview of Evidence) platform, a system that maps questions relevant for health decision-making to a repository of studies developed by the Epistemonikos Foundation and conducts regular updates searches on COVID-19 [50]. We used the health equity typology in development to identify examples. Eligibility criteria were as follows: peer-reviewed COVID-19 observational studies, either descriptive or investigating associations between exposure and health outcomes, which analysed and reported on health equity across any PROGRESS-Plus factor. We chose examples to include different types of studies (e.g., prospective vs. retrospective designs, elements of PROGRESS-Plus, sampling methods, focused and gap approaches, inclusive of high-income vs. low-income countries, and topics). We considered feedback on the examples from our diverse stakeholder team regarding relevance to patients, citizens, clinicians, researchers, decision-makers, and payers of health services and research. We used these examples to inform our proposed interim STROBE-Equity guidance. We circulated the draft guidance with examples to our multidisciplinary, global steering committees on technical oversight, knowledge users, Indigenous research, and citizen and public engagement to engage broader feedback.

## 3. Results

We identified 14 areas in the STROBE checklist that need additional detail to encourage transparent reporting of health equity (Figure 1). Appendix Apresents the full preliminary checklist of possible items for STROBE-Equity reporting guidelines. These items include description of the population across relevant health equity characteristics using the PROGRESS-Plus factors as well as sampling methods to reach and include populations who experience vulnerability. As with CONSORT-Equity, informed consent, research accountability, and ethics procedures need to be reported for all studies that include populations who experience vulnerability and health inequities to promote “never about us, without us”. Studies that include people experiencing inequity need to report methods to determine the relevance of outcomes for these populations and collect relevant socio-demographic and contextual information for analysis. Methods to analyse differential exposure, differential susceptibility and differential capacity to respond need to be planned and described. Finally, implications of exclusion, missingness, or exclusion of people experiencing inequities need to be discussed.

We identified examples of COVID-19 observational studies with an explicit focus on evaluating effects across one or more PROGRESS-Plus factors and summarised these examples of specific questions and transparent reporting of health equity considerations. Table 1 displays examples of focused and gap approaches across PROGRESS-Plus factors in low- and middle-income countries and high-income countries. We defined gap approach as using subgroup analysis or meta-regression to explore differential effects, whereas focused approach examines populations experiencing inequity to assess effectiveness of interventions. The examples also include different types of observational studies detailed in STROBE guidelines. Figure 2 summarises the risk for COVID-19 disease (exposure or susceptibility—i.e., differential effect including infection and recovery) and implications for healthcare access for each PROGRESS-Plus factor.

The components of the PROGRESS-Plus framework are interdependent, and the COVID-19 pandemic has made such interdependence explicit. For instance, South African women have decreased health access compared to men, especially striking among women without post-secondary education [63], or access to the COVID-19 vaccine when scheduling relies on technology is challenging for rural areas and lower-income neighbourhoods within high-income countries [64]. Interactions and intersections of the PROGRESS-Plus factors may amplify disparities (e.g., race and gender, or level of education, incomes, and work conditions), while some may mitigate the effect of others (e.g., social capital). Appendix A provides a comprehensive description of the rationale and relevance to the COVID-19 pandemic and additional examples are also outlined in the Appendix A.

## 4. Discussion

The COVID-19 pandemic has aggravated pre-existing inequities and their effects on health outcomes and societal burden. This interim guidance proposes a structure for applying an equity lens to observational studies by using the well-established PROGRESS-Plus framework. The practical examples strive to illustrate equity considerations in clinical and public health studies and ultimately guide key stakeholders interested in mitigating health inequities. We identified 14 items across six domains of the STROBE checklist that need to be adapted to ensure transparent reporting of health equity.

Western-oriented knowledge and approaches have often reproduced the dynamics of structural oppression systems (racism, colonialism, patriarchy, capitalism, etc.) [65,66]. Thus, the evidence derived from these methodologies is at risk of disregarding the needs of health care and values and preferences of most of the population that, in turn, is composed of a diverse mosaic of groups experiencing multiple axes of disadvantage [67,68,69]. Addressing the social determinants of health as root causes of COVID-19 inequities also involves health professional education, both in undergraduate and postgraduate stages, which tends to explain the complex health–disease processes as mainly relying on the biomedicine model. Medical conferences also need to integrate into the discussion of biomedical information the etiopathogenic role and clinical and social consequences of social determinants of health.

Over the last decades, a number of frameworks have advanced a better understanding of differential health outcomes [48,70,71], highlighting the role of research in identifying and addressing inequity [22,72]. Routine consideration of drivers of health inequities contributes to generating evidence for accountability in evolving political, social, and economic challenges and helps policymakers prioritise actions to maximize both overall population health and the distribution of health in the population [73]. Moreover, frameworks outlining unintended and inequitably distributed harmful impacts may facilitate the development of mitigating strategies [30]. The integration of equity issues requires creative approaches to capture different social dimensions and develop transformative insights [71,74]. For example, training in the conduct of research with Indigenous communities, and drawing attention to who is or is not involved in the research process, including use of the GRIPP (Guidance for Reporting Involvement of Patients and Public) [75] checklist or SAGER guideline [48] required by journals. Broadening research communities and involving more diverse voices may contribute to integrating equity in health research and subsequent decision-making [76,77]. Reporting guidelines have been developed to assist researchers who conduct systematic reviews and RCTs in identifying, extracting, and interpreting evidence on equity [47,49]. Given the fact that the impacts of the COVID-19 pandemic are unevenly and unjustly distributed across groups, it may be critical to appraise equity considerations in any observational investigation whose assessed phenomena might influence health outcomes (e.g., examining unconscious bias regarding racism among clinicians). The STROBE-Equity project is engaged in a comprehensive process to enhance transparency and completeness of reporting of equity considerations in observational studies. Acknowledging that this task requires engaging multiple stakeholders, we call upon interested clinicians, researchers, editors, funding agencies, the public, and policy-makers to join by contacting us through our Open Science Framework project (https://osf.io/h57se/) (accessed on 22 June 2021) [38].

## 5. Conclusions

Given the magnitude of inequities in health and the societal burden of COVID-19, applying an equity lens to observational research can contribute to a better understanding of who may experience vulnerability, including exposure to the virus, response to treatment, community effectiveness, and the consequences of public health policies and other measures. We propose this interim guidance as a structure for integrating an equity lens within the design, analysis, and interpretation and reporting of results in observational research. We identified 14 areas in the STROBE checklist that need additional detail to encourage transparent reporting of health equity; we also described examples that analysed and reported on health equity across social determinants of health. This interim guidance serves as a starting point for clinicians, researchers, decision-makers, and funders in considering what needs to be reported to close the inequitable gaps in health outcomes. The consensus process that is planned for STROBE-Equity extension will further develop parsimonious, evidence-based guidance.

## Figures and Tables

**Figure 1 ijerph-18-09357-f001:**
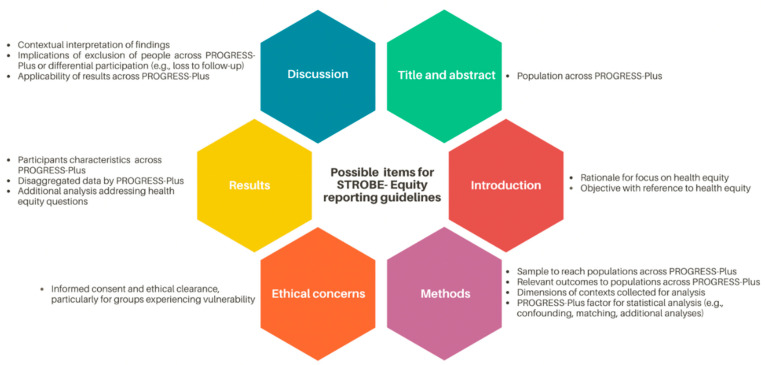
Possible items for STROBE- Equity reporting guidelines. Abbreviations: PROGRESS-Plus, Place of residence, Race/ethnicity/culture/language, Occupation, Gender or sex, Religion, Education, Socioeconomic status, Social capital, and other contextual factors that facilitate disadvantage.

**Figure 2 ijerph-18-09357-f002:**
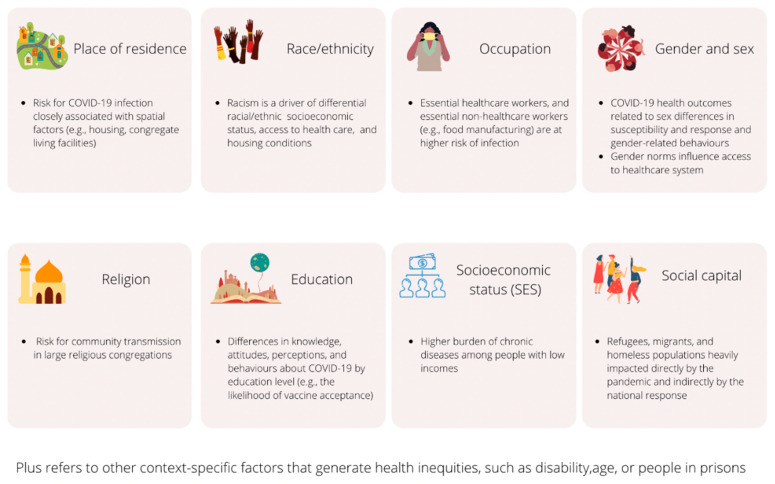
Risk for COVID-19 infection and implications for healthcare access across PROGRESS-Plus factors.

**Table 1 ijerph-18-09357-t001:** Examples of COVID-19 observational studies.

PROGRESS-Plus Factors	Level of Analysis	Context Country Data Source	Gap or Focus Approaches *	Topic	Differences in Health Opportunities	Research Question	Study Design
**Place of residence**	Group	United States Zip code tabulation area	Gap	Incidence Testing access	Differential exposure and susceptibility	What is the characterisation of spatial inequities in COVID-19 testing, positivity, confirmed cases, and mortality during the first 6 months of the pandemic? [51]	Ecological
**Race, ethnicity,** **culture, or language**	Individual	England Hospital and primary care health records	Gap	Prognosis	Differential susceptibility	Are there differences in risk of hospitalisation with severe COVID-19 and/or in-hospital mortality across ethnic groups? [52]	Mixed: cohort and case-control
**Occupation**	Individual	Italy *Ad hoc* database	Focus	Infection prevention and control	Differential exposure	What is the effectiveness of an integrated infection control surveillance system among healthcare workers involved in the first management of suspected or confirmed COVID-19 patients? [53]	Prospective cohort
**Gender**	Group	19 countries from Africa, Asia, Europe, North America, and South America Country-level data	Gap	Vaccination	Differential acceptability	Are demographics, including gender, independently associated with vaccine acceptance? [54]	Cross-sectional
**Sex**	Individual	Sweden, United Kingdon, United States	Gap	Prognosis	Differential susceptibility	What are symptoms associated with short and long COVID adjusting for sex and age? [55]	Prospective cohort
**Religion**	Individual	Ethiopia City-level urban, community-based	Gap	Knowledge, attitudes, perceptions	Differential acceptability	What are factors associated with perception toward COVID-19? [56]	Cross-sectional
**Education**	Individual	Italy City-level Hospital health records	Gap	Prevalence	Differential exposure	What are sociodemographic characteristics, including education level, associated with COVID-19 disease among pregnant females admitted to hospital for pregnancy health care? [57]	Cross-sectional
**Socioeconomic status**	Individual	Peru City-level, Population-based survey	Gap	Prevalence	Differential exposure	What is the seroprevalence of SARS-CoV-2 antibodies in Lima [stratified by socioeconomic status and other variables]? [58]	Cross-sectional
**Social capital**	Group	France Population-based survey	Gap	Policy impact	Differential community effectiveness	What is the impact of lockdown policies on different social classes in France? [59]	Cross-sectional
**Plus**	Individual Children	Spain	Focus	Policy impact	Differential exposure and equity distribution effects	Are social determinants of children’s health unequally distributed during the COVID-19 lockdown? [60]	Cross-sectional
**Indigenous Peoples**	Individual	Brazil Nationwide register	Focus	Incidence	Differential exposure and accessibility to healthcare	What is the burden of COVID-19 on the Indigenous population? [61]	Cohort

Abbreviations: SARS-CoV-2, severe acute respiratory syndrome coronavirus 2. * Definition provided by Welch and colleagues [62]; gap: study evaluates inequities between groups (e.g., across gender), focus: study evaluates inequities only in a group experiencing vulnerability (e.g., essential workers).

## Data Availability

No new data were created or analysed in this study. Data sharing is not applicable to this article.

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
