# Peer review of "Improving Social Justice in COVID-19 Health Research: Interim Guidelines for Reporting Health Equity in Observational Studies"

_ijerph, 2021, doi:10.3390/ijerph18179357_

Round 1

Reviewer 1 Report

Summary and comments for editors and authors:

The authors who are the team members of the STROBE-Equity project draw the reader’s attention to their project which was initiated in January 2021 continuing to 2024. They introduce this project and its aims, call the interested parties to join, and present preliminary findings in a guideline format. The interim guideline is based on the PROGRESS-Plus equity framework which looks at Cov-19 observational studies with the equity lens to encourage future observational health studies to consider, specify and report equity aspects in their methods, ethics, and outcomes.

In summary, the manuscript provides a good equity guideline for future observational studies. The authors should reduce the text by shortening the introduction and part of the results (lines 17-167), and add further information on the used checklists with respective extension in the methods section.

My detailed views/comments on the full text:

Title

It matches the aim, findings, and general content.

Abstract

Is comprehensive and fits the content.

Introduction

Strengths: Is well written, informative, and supports the study by providing relevant information, which convinces the readers of the worth and significance of the study. The aim of the study and the research question both are clear.

Weakness: Too long including many references.

Methods

To develop the interim guideline, this study uses the methods and items below:

·        Holding weekly meetings with the indigenous health experts

·        Considering the PROGRESS-Plus equity factors which are Place, Race, Occupation, Gender, Religion, Education, Socioeconomic status, Social capital, and other contextual factors that facilitate disadvantage

·        Reviewing CONSORT Equity Extension (Consolidated Standards of Reporting Trials)

·        Questioning a citizen panel who were affected by health inequities

·        Searching for the examples of Cov-19 observational studies across one or more PROGRESS-Plus equity factors that were registered in LOVE (Living Overview of Evidence)

·        Circulating the guideline draft for further feedback

Strengths: Introduces important applicable equity-related checklists and extensions and platforms for registering observational studies.

Weakness: At the first glance, it is difficult to recognize/understand quickly some of the mentioned items because only short descriptions and explanations of the used checklists and extensions are provided. More information about the used items would be helpful.

Results

Strengths: Adds equity-related details/suggestions in 14 areas to the existing STROBE checklist (Figure 1) which are significant points for future equity-based Cov-19 observational studies. Besides, it discusses the relations between each PROGRESS-Plus equity factor and Cov-19 by bringing and organizing some studies as an example (Table 1 and Figure 2).

Weakness:

In the main text, the authors provide long descriptions and extra examples for each PROGRESS-Plus equity factor which stretches the paper and reduces the reader’s interest. In my opinion, the examples that are provided in Table 1 and Figure 2 would suffice and there is no need for additional information (lines 17-167) which also leads to too many references (see above). This information can instead come in the appendix.

Discussion

&

Conclusion

Strengths: Well written and conclusive.

References

Strengths: They are relevant to the content, up to date, precise and complete.

Weakness: Too many in main text.

Reviewer 2 Report

Thank you for the opportunity to review this manuscript, which addresses a very important issue and makes a significant contribution to better equity practices. Overall I found the article to be clearly written, however some more specific detail would be helpful in places (please see the comments below).

Abstract

  • The use of these guidelines may improve accountability and social justice - that's a long bow to draw as there are many steps between this resource and those outcomes. I think you could either delete or more closely focus this statement.

  • Final sentence - ? missing word between 'consider' and 'this'

Introduction

  • Final sentence, pg 2 - ? the capability of respond (for both individuals) ... not clear what they are responding to.

  • 1st para, pg. 3 - " due to a lack of agreed upon criteria" - poor grammar in this sentence.

  • 3rd para, pg. 3 - "furthermore, observational studies are well suited" - explain why this is, what is it about this method that makes it particularly appropriate.

  • Consider moving the final paragraph in the intro up, and place it after para3, pg 3.

  • Provide more specific detail about how you engaged with your various stakeholders, and which specific method of co-production you used. This is important to the rigour of your project, and needs to be as replicable as possible.

Materials and Methods

  • Why were the additional modifications made to Moher et al – please provide justification

  • Para 1, final sentence – This seems to be the final stage of the overall project, but its relationship to the data here isn’t clearly linked.

  • How often did the citizen panel meet, and how specifically were they integrated into the development process

  • What were the criteria for choosing examples? What specific factors were considered to make them ‘relevant’, and how was the balance of examples formulated?

Results

  • 1st para – the reporting of equity issues in research doesn’t necessarily ensure ‘nothing about us without us’ – I think this is over-reach.

  • Text on Figure 1 is too small to read comfortably

  • Why did you focus on COVID infection only? Was there a consideration of COVID recovery, where equity issues are increasingly coming to the fore?

  • Text on Figure 2 is also very small, and the brevity of the examples given make it a little simplistic. Doesn’t necessarily add anything to the subsequent discussion of more examples.

    • A good reference to support the conceptualisation of risks is O’Sullivan & Phillips (https://link.springer.com/article/10.1007/s11069-019-03584-6) which talks about the difference between being a ‘medical’ risk and being at ‘functional’ risk.

  • Can you give a specific finding that speaks to the raised morbidity and mortality associated with COVID and patients of culturally diverse communities?

  • How many workers experienced differential vulnerability [74] – money is a bit imprecise.

  • ‘Segregation is mainly drive by religion’ – do you mean healthcare segregation or something else?

  • Can you give specific examples of self protective behaviours during the pandemic [97]?

  • ‘measured by deconstructing into bonding’ – poor grammar in this sentence

  • ‘social capital results in tangible disparities’ – can you provide specific examples of the relationship between social capital and equity in relation to COVID?

  • Indigenous peoples – Final two sentences are very general and could be construed as ‘motherhood’ statements. Can these be related to more specific issues to anchor it in the experiences of indigenous peoples?

  • Which chronic diseases have been found to be predictive of COVID severity?

  • Final sentence in the multimorbidity section – poor grammar.

Discussion

  • It's not just medical staff who could benefit from this education – please consider saying health professional education or another more inclusive term

  • The link to the GRIPP Guidelines was great – really shows how this can integrate with other guidelines that researchers may be referring to. Are there other examples of these relationships you could cite? OR could this be a diagram showing the place of these guidelines in relation to other guidelines / checklists?

  • You’ve made a call to join you, but how do interested people do that?

Conclusion

  • Second last sentence – poor grammar
